# Pathologically Decreased CSF Levels of Synaptic Marker NPTX2 in DLB Are Correlated with Levels of Alpha-Synuclein and VGF

**DOI:** 10.3390/cells10010038

**Published:** 2020-12-29

**Authors:** Walter A. Boiten, Inger van Steenoven, Mei-Fang Xiao, Paul F. Worley, Barbara Noli, Cristina Cocco, Gian-Luca Ferri, Afina W. Lemstra, Charlotte E. Teunissen

**Affiliations:** 1Neurochemistry Department, Clinical Chemistry, Amsterdam UMC, 1081 HV Amsterdam, The Netherlands; i.vansteenoven@amsterdamUMC.nl (I.v.S.); c.teunissen@amsterdamumc.nl (C.E.T.); 2Department of Neurology, Alzheimer Center Amsterdam, Amsterdam UMC, 1081 HV Amsterdam, The Netherlands; a.lemstra@amsterdamumc.nl; 3Department of Neuroscience, Johns Hopkins University School of Medicine, Baltimore, MD 21205, USA; mxiao1@jhmi.edu (M.-F.X.); pworley1@jhmi.edu (P.F.W.); 4NEF-Laboratory, Department of Biomedical Sciences, University of Cagliari, 09042 Monserrato, Italy; barbaranoli@yahoo.it (B.N.); crcocco@unica.it (C.C.); ferri@unica.it (G.-L.F.)

**Keywords:** dementia with Lewy bodies, Alzheimer’s disease, neuronal pentraxin 2, VGF (non-acronym), α-synuclein, biomarkers, cerebral spinal fluid, cognitive function, visual spatial domain

## Abstract

**Background:** Dementia with Lewy bodies (DLB) is a neurodegenerative disease where synaptic loss and reduced synaptic integrity are important neuropathological substrates. Neuronal Pentraxin 2(NPTX2) is a synaptic protein that drives the GABAergic inhibitory circuit. Our aim was to examine if NPTX2 cerebral spinal fluid (CSF) levels in DLB patients were altered and how these levels related to other synaptic protein levels and to cognitive function and decline. **Methods:** NPTX2, VGF, and α-synuclein levels were determined in CSF of cognitive healthy (*n* = 27), DLB (*n* = 48), and AD (*n* = 20) subjects. Multiple cognitive domains were tested, and data were compared using linear models. **Results:** Decreased NPTX2 levels were observed in DLB (median = 474) and AD (median = 453) compared to cognitive healthy subjects (median = 773). Strong correlations between NPTX2, VGF, and α-synuclein were observed dependent on diagnosis. Combined, these markers had a high differentiating power between DLB and cognitive healthy subjects (AUC = 0.944). Clinically, NPTX2 levels related to global cognitive function and cognitive decline in the visual spatial domain. **Conclusion:** NPTX2 CSF levels were reduced in DLB and closely correlated to decreased VGF and α-synuclein CSF levels. CSF NPTX2 levels in DLB related to decreased functioning in the visual spatial domain.

## 1. Introduction

Examining the different types of neurodegenerative processes in living individuals is challenging, as multiple pathological processes are concurring in the aging brain leading to comparable clinical presentations [1,2]. To better understand the disease trajectory and how the underlying biological mechanisms affect disease progression, it is essential to accurately measure these processes during a individuals lifetime. In dementia with Lewy bodies (DLB), multiple pathological processes converge. One is the abnormal accumulation of α-synuclein into Lewy bodies (LBs) that is also observed in Parkinson disease (PD), multiple systems atrophy, and Parkinson’s disease dementia (PDD). Second, in a substantial portion of DLB patients (40–60%), pathological changes of Alzheimer’s disease (AD) are also observed (Aβ_1–42_ aggregation into amyloid plaques and the phosphorylation and aggregation of tau protein) [3]. Last, reduced synaptic integrity and synaptic loss are integral pathological processes occurring in both DLB and AD [4,5]. Specific synaptic loss has been observed in primary visual cortex of DLB patient [6].

Cognitive decline in DLB differs from that in AD, since it shows a more pronounced effect on the visual spatial, attention, and executive function domains, in contrast to the predominant effects on the memory domain in AD [7,8,9]. In areas related to these domains, synaptic dysfunction might play an essential role in this clinical presentation [10]. Furthermore, synaptic dysfunction and loss in DLB have been examined using post-mortem material, and these studies underlined its potential role in disease progression [4,6,11,12]. Studying synaptic integrity in the progression of DLB can therefore give a better understanding of the underlying pathological process. Yet, acquiring material for direct studies of synaptic integrity in living subjects is not feasible and therefore synaptic biomarkers will be a good alternative.

Fluid biomarkers can be a powerful tool to study the disease processes occurring in the aging brain of living individuals. This is illustrated by their use in AD, where abnormal cerebral spinal fluid (CSF) level of Aβ_1–42,_ phosphorylated tau and total tau aid diagnosis [13,14] and can be used to outline the disease trajectory [13]. In a recent CSF proteomics study, we identified novel potential DLB biomarkers, including neuronal Pentraxin 2 (NPTX2), VGF (a non-acronym), and secretogranin II (SCGII) [15]. Interestingly, these markers are all related to synaptic functioning and synaptic vesicles. These markers have also been observed in a recent AD proteomics study [16], indicating their presence in CSF in relation to a more general neurodegenerative phenotype.

NPTX2 is a secretory synaptic protein expressed pre-synaptically by principle neurons and released at excitatory synapses connected to parvalbumin interneurons [17]. As an immediate early gene, NPTX2 has an important role in activity dependent synaptic plasticity. Its expression can be induced by neuronal activity and brain derived neurotrophic factor (BDNF) [18]. NPTX2 forms a complex with two other pentraxins: NPTX1 and NPTXR [19]. After exocytosis, this complex is able to mediate AMPA receptor, clustering post-synaptically at GABAergic interneurons [20]. This makes it a selective marker for the principle neuron–interneuron inhibitory circuit [21]. Positive correlations of NPTX2 with other synaptic markers like neurogranin and SNAP-25 were observed, although weaker than between these markers themselves, due to NPTX2’s selectivity [22]. Yet, we hypothesize that synaptic proteins also under control of BDNF might show similar alterations. Previous studies have shown a decreased CSF level of NPTX2 in AD patients, FTD, Down syndrome patients, CSF, and neuronal derived extracellular vesicle of AD patients [21,22,23,24,25]. These levels were related to cognitive decline and functional connectivity, but these relations of NPTX2 in DLB have not yet been examined.

The other candidate biomarker in the DLB proteomics study, VGF, is an interesting marker to compare to NPTX2, as its transcription is induced by neuronal growth factor (NGF) and BDNF [26,27]. This BDNF pathway is impacted in neurodegeneration, and VGF was implicated as a master regulator for NPTX2 [28]. VGF peptides are stored in dense core vesicles (DCVs), function as secretory peptide at synapses, and are important for a number of neuronal functions [27,29]. A detailed study into the role of VGF peptide GGEE45 has shown decreased CSF levels in DLB patients and levels correlated to cognitive decline in multiple cognitive domains [30]. VGF has also been implicated as an AD biomarker and changes in relation to disease progression [28,31]. Therefore, the relation between VGF and NPTX2 levels in CSF might give insight into changes in VGF/BDNF driven synaptic functioning and dysfunction under pathological conditions.

Both VGF and NPTX2 are related to synaptic function and are secretory proteins, which associates them mechanistically to the main pathological protein in Lewy Bodies: α-synuclein. Although the exact physiological function of α-synuclein is not clearly understood, it is strongly suggested that it functions as a lipid binding protein interacting with DCVs at the presynaptic site and plays a role in regulating exo- and endocytosis and vesicle recycling [32,33,34]. The pathological role of α-synuclein in DLB has also been linked to presynaptic functioning, where small α-synuclein aggregates in the presynaptic neuron caused neurodegeneration [12].

We hypothesize that under physiological conditions, synaptic functioning, and thus levels of these markers, are related. Therefore, changes in one might be reflected in the other. In contrast, under pathological conditions, synaptic dysfunction and loss will occur reducing the actively released synaptic protein levels (like NPTX2 and VGF) and might relatively increase vesicle endocytosis related synaptic protein (like α-synuclein). Changes in CSF levels of α-synuclein have been reported in both AD and DLB [35,36], where both increased and decreased levels of α-synuclein have reported in DLB patients CSF [37,38]. Yet, large variations in levels of α-synuclein were observed within groups indicative of a large biological variation. We examined the CSF levels of NTPX2 from DLB patients and compared them to levels of VGF and α-synuclein. A comparable decrease to that of AD patients was observed in DLB patients when compared to cognitively healthy subjects. To examined if changes in synaptic markers were related, correlations between NTPX2, VGF and α-synuclein in cognitively healthy subjects and disease were used. To study if these synaptic functions were reflected clinically, NPTX2 levels were related to cognitive test scores. These findings showed that synaptic dysfunction and loss in DLB is reflected in CSF biomarkers, and a principal neuron-interneuron inhibitory circuit specific marker NPTX2 was related to decline in visuals spatial task performance.

## 2. Materials and Methods

### 2.1. Ethics, Subject Selection, and Description

In this study, a total of 95 subjects were included from the Amsterdam dementia cohort [39], 27 with subjective cognitive decline (SCD), 48 with DLB, and 20 with AD. The diagnosis AD was established using the National Institute on Aging and Alzheimer’s Association (NIA-AA) criteria and was CSF biomarker confirmed (Aβ_1–42_, total tau, and phosphorylated-tau (181)). DLB patients were diagnosed using the international diagnostic consensus criteria for probable DLB [9], and diagnosis was supported by (123) I-FP-CIT-SPECT (DAT-SPEC) scan for 41 DLB patients. Subjects were labelled as SCD when they presented to the memory clinic with subjective memory complains, yet had an unimpaired performance levels on the cognitive test, and criteria for mild cognitive impairment, AD, or other neurological or psychiatric disorders were not met [40]. SCD subjects were defined as the cognitively healthy control group. A description of the study population is given in Table 1. Group sizes are indicated with each specific test performed.

All subjects in this study underwent a standardized multidisciplinary examination including medical history, physical and neurological conditions, neuropsychological testing, electroencephalography, brain magnetic resonance imaging, and a lumbar puncture for biomarker testing. Thereafter, subjects were diagnosed in a multidisciplinary consensus meeting. This study was performed according to the declaration of Helsinki and its subsequent amendments, approved by the VUmc medical ethical committee, and all subjects signed an informed consent form specifying that clinical data and CSF can be used for research purposes.

### 2.2. CSF Biomarkers

The three CSF AD biomarkers, Aβ_1–42_, total-tau (T-tau), and phosphor-tau (181) (p-tau), were determined using a commercial ELISA assay (Innotest, fujirebio, Gent, Belgium). Levels of Aβ_1–42_ were corrected for the methodological drift occurring over the years [41]. Levels of NPTX2 were determined using an in-house-sandwich ELISA preformed as previously described in [21]. Briefly, the ELISA was based on a rabbit anti-NPTX2 capture and mouse anti-NPTX2 detection antibody. To determine the levels of VGF, a competitive ELISA was used with an antibody raised against amino acid 373–417 in of the VGF protein in rabbits, as previously described [30]. For the determination of α-synuclein level, an ELISA from ADx neurosciences (Gent, Belgium), available through Euroimmun, was used [42].

### 2.3. Neuropsychological Examination

To determine the cognitive performance on multiple memory domains, a number of standardized neuropsychological tests were performed. Each domain encompassed a number of tests. As a general cognitive test, the Mini mental state exam (MMSE) was performed. The following cognitive domains were investigated and the test used for each domain are listed:•Memory domain:○Rays auditory verbal learning test (RAVLT) immediate recall○RAVLT delayed recall○Visual association test A was used•Attention○Digit span forward○Trail making test A○Stroop 1○Stroop 2•Executive function○Digit span backward○Stroop 3○Trail making test B○Letter fluency○Frontal assessment battery•Language○Dutch version of the Boston naming test○Category fluency test•Visual spatial○Visual object and space perception (VOSP) number location○VOSP dot counting○VOSP frat let

To calculate each of the cognitive domain scores, the underlying test scores were first z-scored using a population average and standard deviation (Appendix A) and if necessary inverted, after which the average of these z-scores was taken only when more than one test score was available. A global cognitive score was calculated using the median of all z-scored test results if more than four scores were available. The median was selected, as it gives a more comparable approximation of a composite score when an unequal number of individual test scores are available (e.g., an outlying value will add less). For 33 DLB patients, follow-up data were available and used to study cognitive decline over time.

### 2.4. Statistics

Groups were compared using Kruskal–Wallis, as most levels were non-normally distributed (Appendix A depict the results of normality tests) with a Bonferroni corrected post-hoc test. To normalize the distributions, the levels of the three biomarkers were Ln transformed (Appendix A). Linear models, binary logistic, and mixed models used these transformed data. Since DLB is more frequent in males, a larger number of males could be included in this study. Yet, percentage of male subjects was kept similar across groups. Although sex differences can have a relevant impact on biomarker levels in DLB [43], they were not included as a parameter in our models, due to the limited power for most study groups. All statistics were performed using IBM^®^ SPSS statistics version 26 (Armonk, NY, USA).

To determine correlations between NTPX2 levels and cognitive function in DBL, linear models were used with Ln (NPTX2) level, age, and education as covariates. To determine the relations to cognitive decline, mixed models with random intercept and time, age, and education as covariates and cognitive function as dependent variable were used. An interaction term of the Ln (NPTX2) level with time was included to determine if NTPX2 levels related to changes in cognitive score over time. For Receiver Operator Curves (ROC) analysis, binary logistic regression was performed using age as a covariate and the Ln transformed marker data. The combined model included all three markers in the same regression model.

## 3. Results

### 3.1. Study Population

Basic characteristics of the studied population are given in Table 1. There, the mean (SD) of the z-scored cognitive domain scores and median with a range of 95% of the biomarker levels are also given. In DLB, lower levels of Aβ_1–42_ and higher levels of t-tau were observed, but no significant changes in p-Tau levels. The cognitive scores between the three groups showed significantly lower scores for both the DLB and AD groups, where the AD group had a significantly lower MMSE than the DLB group. A similar decrease in AD and DLB compared to controls was observed for the AD biomarker data. Differences in CSF levels of NPTX2 compared to SCD were observed for DLB and AD (Figure 1A depicts the levels of NTPX2). As previously presented, CSF VGF levels were only significantly decreased in DLB and not in AD [30], and the CSF levels of α-synuclein were only increased in the AD group (Appendix A). Correcting for age gave comparable results except for VGF levels, which differed between AD and DLB (Appendix A depicts the models with age). No consistent changes in these markers between groups were observed.

### 3.2. Correlation to Other Synaptic Markers

To examine if there was a relation between NPTX2, VGF, and α-synuclein levels in CSF, and if this relation was diagnosis dependent, the transformed data were plotted against one another dependent on diagnosis (Figure 1B and Appendix A). The Pearson r and Spearman R values between the three markers were calculated for all diagnostic groups combined and for each independently (Figure 1C and Appendix A). Correlations of levels for the three biomarkers moved in similar directions, which was most apparent in the SCD group. Thus, in cognitively healthy subjects, CSF levels of one marker are reflected in levels of the other two markers, and this size of this relationship is maintained between different subjects. In DLB, correlations between markers still moved in similar directions as they did in SCD; however, a deviation seen as an offset from the relation observed in SCD can be observed. Appendix A gives an estimation of this offset focusing on α-synuclein levels, and illustrates significant difference between SCD and DLB. In AD, correlations to NPTX2 were diminished, yet the correlation between VGF and α-synuclein remained. Correlations of NPTX2 to other AD biomarkers in the DLB group were numerically less than those to VGF and α-synuclein (Figure 1C).

### 3.3. Differentiating Potential of Combined Markers

Although levels of NPTX2 are significantly reduced in the DLB diagnostic group, they are expected not to be a strong predictor, due to the large variation and overlap between DLB and SCD groups. However, when examining the levels in relation to two other related synaptic markers (VGF and α-synuclein), a clear distinction was observed (the offset between SCD and DLB in Figure 1B). To determine the differentiating potential, binary logistic regression with corresponding ROC curve analyses were performed for each marker independently and for all markers combined. All models were corrected for age, because age on its own proved to be as strong a predictor. Figure 1D depicts the ROC curve and corresponding AUCs. The combination of makers showed a better sensitivity and specificity than any of the markers separately, also reflected in the AUCs. Of the three markers, α-synuclein showed no added value to age. VGF and NPTX2 showed comparable low AUCs.

Similar analyses were performed comparing AD to SCD and AD to DLB. Binary logistic regression models were run for each marker independently and all markers combined, including age as covariate in all analysis. ROC curve analysis was performed for both comparisons and is shown in Appendix A. The model with all three markers combined had the highest sensitivity and specificity differentiating AD and SCD (combined markers AUC = 0.963). This was mostly driven by NPTX2 (AUCS = 0.795) and α-synuclein (AUC = 0.782) levels, which was reflected in their individual ROC curves being the only two significantly different from 0.5. Limited sensitivity and specificity were observed for these markers when differentiating between AD and DLB (combined markers AUC = 0.759).

### 3.4. NPTX2 Levels Relate to Cognitive Function and Decline

As DLB patients have reduced performance in multiple cognitive functional domains and NPTX2 levels could indicate dysfunction in specific inhibitory interneuronal circuits, the relation of NPTX2 with these domains was examined using linear models. Table 2 depicts the beta, significance level and 95% confidence intervals of the relation between cognitive function and CSF NPTX2 levels of DLB patients. A significant positive correlation to the median global cognitive score was observed, yet not for the MMSE. When the separate domain scores were examined, a significant relation with the attention domain and a substantial trend toward decline in the language domain were observed.

We next studied the relation of NPTX2 towards cognitive decline. A significant relation was observed for the NPTX2 level at LP to subsequent decline of the visual spatial domain. None of the other declines in scores showed a relation or trend towards the CSF NPTX2 level. All cognitive domains showed significant decline over time, when modelled without NPTX2 interaction. When including NPTX2, the beta represents the modulating effect that NPTX2 levels have on the time parameter (interaction NPTX2 CSF level*time). A positive beta would relate decreased levels to steeper decline in cognitive score (Table 2). Relations between NPTX2 CSF levels and cognitive function and decline were examined for AD patients using similar models as applied for the DLB patients. Data are shown in Appendix A. No significant relations between NPTX2 level and cognitive function or decline were observed in the AD group. For both the MMSE and global cognitive score, 95% confidence intervals indicate a possible positive relation between NPTX2 levels and the cognition; however, these were not significant due to limited power in this group.

## 4. Discussion

Here, we have examined the levels of the synaptic marker NPTX2, related these to levels of VGF and α-synuclein in DLB, and compared these to their relation in SCD and AD. Combined these markers gave an increased differentiating potential of DLB and SCD. Furthermore, we show that in DLB, levels of NPTX2 were related to cognitive function and decline in the visual spatial domain score. Differences between results for a relation of NPTX2 with MMSE and global cognitive domain scores suggest that the global score might be more sensitive to NPTX2 related synaptic loss in domains specifically affected in DLB. The fact that NPTX2 showed only relations to a few specific domain scores might be due to limited subject numbers. However, this might also be due to variation in NPTX2 CSF levels not directly translating to neuronal NPTX2 levels and representing biological variation. Previously, similar findings relating NPTX2 to cognitive decline had been observed in AD [21,22]. These findings were not observed in the current study, which was probably due to limited number of AD patients compared to these previous studies, especially for the evaluation of cognitive decline during follow-up. However, a trend for the relation between the MMSE and global cognitive score was observed. With the data presented here, we show that NPTX2 and synaptic biomarkers can be potent tool to study neurodegenerative diseases, which we will further delineate below.

In the data presented, we observed reduced NPTX2 levels in CSF of DLB patients, comparable to those observed in AD. NPTX2 functions as signal regulator in the excitatory synapses between principal neuron and GABAergic interneurons [17], and activity and number of these synaptic connections would impact NPTX2 levels in CSF. Lower NPTX2 levels would thus be an indication of lower excitation of inhibitory neuronal networks and thereby decreased GABAergic inhibitory signaling. This process has been related to memory decline and hippocampal volume in AD [21], and the changes observed in DLB might also relate to these processes. However, memory and hippocampal volume are less prone to suffering from degeneration in DLB [2,9]. We observed a prominent relation between NPTX2 levels and visual spatial decline. Interestingly, the GABAergic interneurons of the visual cortex have been implicated in DLB [6], and decline of synaptic functions in this region is also observed in DLB [4]. Together, these findings suggest that reduced levels of NPTX2 might play a critical role in visual problems commonly observed in DLB patients by interfering in the principal neuron-interneuron circuitry. Yet, further validation studies of NPTX2 expression in post-mortem tissue of DLB patients and validation in another DLB cohort would be required to confirm this hypothesis.

Why are levels of NPTX2 decreased in DLB patients? Drivers behind lower NPTX2 values might be neurodegeneration and concomitant synaptic loss. Yet, a close correlation between NPTX2 and VGF was observed in cognitively healthy subjects, and this was maintained in both DLB and in AD, although somewhat weaker. Recently, VGF has been implicated as a master regulator of NPTX2 and BDNF [28], where BDNF itself also functions as regulator of both NPTX2 and VGF [18,26]. In this process, BDNF and VGF are seen as key regulators of neurogenesis, synaptic plasticity, and cognition [28]. It can be hypothesized that NPTX2 is under regulation of the VGF/BDNF pathway. This would explain the close relation of the selective marker NPTX2 (a marker of a more selective process) to the more generally expressed marker VGF. Unfortunately, BDNF levels were not included in this study. Yet, lower levels of VGF and BDNF might be causal to the decreased levels of NPTX2. Monitoring multiple proteins that can be related by function, as in this study, can provide a measure of how this function changes during disease pathogenesis.

The closest relation of NPTX2 or VGF to known CSF markers was observed for α-synuclein (stronger than for Aβ_1–42_ and tau). VGF and α-synuclein might be mechanistically linked, as VGF is excreted by DCVs and α-synuclein functions in DCV recycling [29,33]. In cognitive healthy individuals, a balance between the levels of these proteins in CSF is observed, probably due to their shared biological mechanism. NPTX2 also closely correlated to both in cognitively healthy subjects, as it is also a secretory synaptic protein. In DLB, small aggregates of α-synuclein have been shown to disrupt pre-synaptic functioning [12], thereby possibly reducing the secretion of VGF and other neurotrophic factors, consequently promoting further synaptic loss. Due to more synaptic loss and less synaptic functioning, more of the structural protein α-synuclein may end up in the interstitial fluid as synapses deteriorate, both causing an offset from the physiological processes still occurring. This was clearly observed in DLB, where the normal relations are put at an offset with relative higher α-synuclein levels and lower VGF and NPTX2.

Under physiological conditions, many synaptic proteins show close correlations, as seen for neurogranin and SNAP-25 [21], α-synuclein and neurogranin [44], and Contactin 2 and α-synuclein [45]. These markers could be used to paint a more accurate picture of synaptic function using CSF. Using multiple markers of the same biological process can reduce the biological variation normally observed with such markers. This was clearly observed for VGF, which could explain 79% of the variation in α-synuclein in SCD subjects. Including multiple markers of a similar dysfunction (e.g., synaptic loss) in a diagnostic model increased the sensitivity and specificity of this model, which was observed here when combining VGF, NPTX2, and α-synuclein. Another marker not examined here, which could expand the biological relation, is SCGII. This is a secretory protein also related to DCV and has showed alterations in two proteomics studies where VGF and NPTX2 were also implicated [12,13]. SCGII might be an interesting target for further studies of synaptic function in DLB. Understanding how relations between synaptic markers change over the disease course might give more potent insight into the pathological process and etiology. Although counterintuitive, using a cross-sectional approach where multiple markers are compared might give a novel manner of studying disease progression. Deviations from relations between biomarkers observed in cognitive healthy subjects might be indicative of disease severity, as is done in AD with the AT(N) norms [13].

Although CSF levels of NPTX2 were deceased in DLB, similar changes in CSF levels are observed in AD. Therefore, NPTX2 might not be suited for clinical diagnostic purposes or to distinguish different mechanisms causing pathological synaptic dysfunction and loss. Yet, for future mechanistic studies it needs to be considered that these markers could be indicative of different pathological processes specific to either DLB or AD. In the former, we hypothesize here that decreased levels might be considered in relation to GABAergic interneuron of the visual cortex. In the latter, studies have shown a relation between NPTX2 and hippocampal volume and functional connectivity in the salient/attention networks [21,23].

In conclusion, decreased CSF levels of NPTX2 might be indicative of a specific neurodegenerative processes in DLB related to the visual spatial domain. In both DLB and AD, a possible explanation for these decreased levels, based on literature, suggest that this might be driven by decreased signaling via the VGF/BDNF pathway. Further understanding of how these processes intertwine and how they are reflected in CSF could give insight into the neurodegenerative processes occurring in both DLB and AD.

## Figures and Tables

**Figure 1 cells-10-00038-f001:**
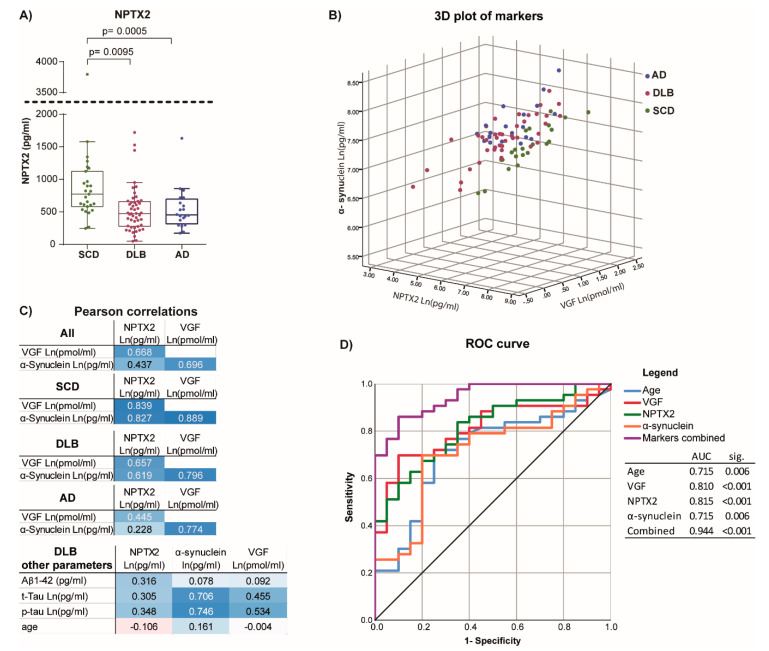
NPTX2 levels and relation to biomarkers in CSF. (**A**): Levels of NPTX2 in CSF of SCD subjects and DLB and AD patients. Box-plots are median and interquartile ranges, whiskers are 95% percentiles. When significant, the *p*-value of Kruskal–Wallis with post-test is indicated. (**B**): 3D plot of the levels of the three related synaptic markers, NPTX2, VGF, α-synuclein. All levels were Ln transformed. (**C**): Pearson r values for correlations between the three transformed marker levels for each diagnostic group independently. In the bottom table, the Pearson r values for the correlation to AD biomarkers and age are given. Color intensity indicates the strength of the correlation, with blue being stronger positive correlations and red stronger negative correlations. (**D**): ROC curve distinguishing between SCD and DLB. For all models, age was included. Markers combined is a model using NPTX2, VGF, and α-synuclein combined. AUCs and a significance value indicating if the significantly deviate for 0.5 are given.

**Table 1 cells-10-00038-t001:** An overview of the studied population. Levels of all parameters used in this study are represented either as mean (SD) or as median with upper and lower 95% confidence intervals. Cognitive scores are given as z-scores except for the MMSE. Levels and numbers of *n* are indicated per diagnostic subgroup. For each subgroup, the *p*-value of the F-test from a Kruskal–Wallis is indicated and when significant, an adjust *p*-value of the post-test. Significant values are indicated by a bold, underlined, and italic font.

	SCD *n* = 27		DLB *n* = 48		AD *n* = 20		*p*-Value			
	Mean (SD)	*n*(%)	Mean (SD)	*n*(%)	Mean (SD)	*n*(%)	F-Test	SCD-DLB	SCD-AD	DLB-AD
Female %		4(15)		6 (13)		2 (10)				
Age	63.7 (5.9)		67.7 (6.4)		65.3 (6.0)		** * 0.018 * **	** * 0.015 * **	0.827	0.560
Education	5.7 (1.0)		5.1 (1.3)		5.3 (1.0)		0.138	N.R.	N.R.	N.R.
Global	−0.21 (0.53)	21	−2.48 (1.16)	40	−4.00 (2.03)	17	** * <0.001 * **	** * <0.001 * **	** * <0.001 * **	0.139
Memory	−0.15 (0.8)	21	−2.76 (1.7)	40	−2.99 (1.95)	13	** * <0.001 * **	** * <0.001 * **	** * <0.001 * **	1.000
Attention	−0.28 (0.71)	21	−2.86 (2.15)	40	−3.63 (2.48)	13	** * <0.001 * **	** * <0.001 * **	** * <0.001 * **	1.000
Executive function	−0.23 (0.73)	21	−2.49 (1.29)	40	−3.08 (1.35)	13	** * <0.001 * **	** * <0.001 * **	** * <0.001 * **	0.792
Language	−0.13 (0.61)	21	−1.14 (0.68)	40	−1.99 (1.74)	12	** * <0.001 * **	** * <0.001 * **	** * <0.001 * **	0.780
Visual spatial	−0.21 (0.89)	14	−2.22 (1.85)	30	−4.79 (3.27)	10	** * <0.001 * **	** * 0.003 * **	** * <0.001 * **	0.166
MMSE	28.12 (2.02)	18	23.07 (4.40)	42	18.67 (542.)	18	** * <0.001 * **	** * <0.001 * **	** * <0.001 * **	** * 0.048 * **
	Median [95%interval], *n*		Median [95%interval], *n*		Median [95%interval], *n*		F-test	SCD-DLB	SCD-AD	DLB-AD
Aβ1–42 (pg/mL)	1032 [982–1112], 26		788 [760–905], 47		586 [531–614], 19		** * <0.001 * **	** * 0.002 * **	** * <0.001 * **	** * <0.001 * **
T-Tau (pg/mL)	199 [182–237], 26		299 [285–393],48		596 [522–949], 19		** * <0.001 * **	** * 0.003 * **	** * <0.001 * **	** * <0.001 * **
p-Tau (pg/mL)	41.0 [34.7–42.8], 26		47.0 [43.4–58.6], 47		87.5 [77.6–120.7], 20		** * <0.001 * **	0.101	** * <0.001 * **	** * <0.001 * **
NPTX2 (pg/mL)	773 [581–1123], 27		474 [279–659], 48		453 [317–696], 20		** * <0.001 * **	** * <0.001 * **	** * 0.009 * **	1.000
α-synuclein (pg/mL)	1508 [1281–1971], 22		1811 [1537–2299], 44		2009 [1764–2658], 19		** * 0.004 * **	0.156	** * 0.003 * **	0.166
VGF (pmol/mL)	3.66 [2.97–5.15], 22		2.58 [2.04–3.42] 44		3.33 [2.40–4.41] 20		** * 0.007 * **	** * 0.009 * **	1.000	0.136

**Table 2 cells-10-00038-t002:** The relation of NPTX2 to cognitive function and cognitive decline in DLB patients.

			95% Confidence	
Cognitive function	β	Sig.	Lower	Upper
Global median *n* = 39	** * 0.2642 * **	** * 0.045 * **	** * 0.006 * **	** * 0.522 * **
Memory *n* = 39	0.1931	0.595	−0.538	0.925
Attention *n* = 39	** * 1.0938 * **	** * 0.019 * **	** * 0.194 * **	** * 1.994 * **
Executive =39	−0.0957	0.755	−0.714	0.523
Language *n* = 39	0.2681	0.063	−0.015	0.552
Visual *n* = 30	−0.7258	0.197	−1.853	0.401
MMSE *n* = 40	0.7791	0.406	−1.102	2.660
Cognitive decline				
Global median *n* = 32	0.0006	0.837	−0.005	0.007
Memory *n* = 32	0.0019	0.831	−0.016	0.020
Attention *n* = 32	0.0056	0.753	−0.030	0.041
Executive *n* = 32	0.0001	0.985	−0.014	0.014
Language *n* = 32	0.0031	0.393	−0.004	0.010
Visual *n* = 28	** * 0.0436 * **	** * 0.002 * **	** * 0.016 * **	** * 0.071 * **
MMSE *n* = 33	0.0014	0.950	−0.041	0.044

For each domain, the beta significance level, and 95% confidence intervals of a model relating NPTX2 to that specific cognitive domain are given. The upper set depicts those values for linear models with cognitive function. The lower set shows the values for a linear mixed model for cognitive decline for each domain. Significant values are indicated by bold underline italic fonts.

## Data Availability

The data presented in this study are available on request from the corresponding author.

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
