# Peer review of "Pathologically Decreased CSF Levels of Synaptic Marker NPTX2 in DLB Are Correlated with Levels of Alpha-Synuclein and VGF"

_cells, 2020, doi:10.3390/cells10010038_

Round 1
Reviewer 1 Report
In this manuscript, Boiten et al., have analysed the levels of NPTX2 and other related synaptic markers in CSF sample from control, DLB and AD patients. They have cross-related their findings with the cognitive performance in several test related to memory, attention executive function etc.
The study is well performed. The cohort well characterised. In fact very well characterised in terms of cognitive function. Also, the statistical analysis is solid and multiple variables and conditions have been addressed. However, my feeling is that the authors failed to support their conclusions with the data reported in the study.
In the abstract section, the authors concluded that: “ NPTX2 CSF levels were reduced in DLB and might be linked to decreased VGF expression. CSF NPTX2 levels in DLB might relate to decreased functioning in the visual spatial domain due to a disturbed GABAergic inhibitory circuit”. Also in the discussion section, the authors wrote that: “… decreased levels of NPTX2 might be a potent biomarker for specific neurodegenerative processes in DLB related to spatial domain………. could give insight into the neurodegenerative processes occurring in DLB and how might differ from AD…”.
Data support that decrease in NPTX2 CSF levels correlate with the reduction in VGF but it is not possible to go further from that. No mechanistic data supports that this is the case. It is also clear that there is a correlation between NPTX2 but it is not possible to infer a this reduction is related to VGF and α-synuclein as is indicated in the title of the manuscript. Moreover, changes observed doesn’t seem to be specific to DLB. Similar reductions are observed in CSF from AD, thus it is difficult to image how these data could give insight into the mechanistically differences that would underlie between DLB and AD. An interesting observation is the eventual relationship between NPTX2 levels in the CSF and cognitive performance in the visual domain. However, I found that it was not a comment onto the differences found in visual domain cognitive function vs cognitive decline.
In general, it si difficult to obtain specific DLB mechanistic insight from CSF values of some proteins with change in other neurodegenerative diseases.
Specific points
- Adjust the conclusions to the data obtained and also change in the title related to correlated.
- In order to be able to compare with AD, it is necessary that the authors include the same analysis showed in Fig 1D (ROC analysis) and Table 2 with the data obtained from AD CSF.
Author Response
Rebuttal addressing the points of the reviewer
We thank the review for the thorough examination of the manuscript and the well-thought-out points of consideration. We hope that we have adequately addresses the points mentioned by the reviewer for acceptance of this manuscript.
Point 1 reviewer: Adjust the conclusions to the data obtained and also change in the title related to correlated.
Response:
We fully agree with the reviewer that CSF levels of NPTX2 are unable to give insight into the mechanistically differences between pathological processes which can both lead to decreased NPTX2 levels. The absence of this distinction limits the use of NPTX2 as a biomarker for clinical diagnostics between DLB and AD. It illustrates the communalities between pathological synaptic dysfunction and loss occurring in either disease, however, it does not exclude that these might be dissimilar in spatial origin and etiology. To that end, we have changed multiple parts of the manuscript as suggested by the reviewer. In bold the added change and strikethrough is used for deleted parts.
Title:
“Pathologically decreased CSF NPTX2 levels in DLB correlate are related to the synaptic marker’s alpha-synuclein and VGF”
Abstract: lines 27-30
“Conclusion: NPTX2 CSF levels were reduced in DLB and closely correlated might be linked to decreased VGF and α-synuclein CSF levels. CSF NPTX2 levels in DLB might related to decreased functioning in the visual spatial domain due to a disturbed GABAergic inhibitory circuit.”
Introduction: Lines 107-110
“These findings showed that synaptic dysfunction and loss in DLB is reflected in CSF biomarkers and a the principal neuron-interneuron inhibitory circuit specific marker NPTX2 was related to decline in visuals spatial task performance. specific loss of the principal neuron-interneuron inhibitory circuit might underlie functional deficits in the visual cortex.”
Discussion: lines 358- 372
“Although CSF levels of NPTX2 were deceased in DLB, similar changes in CSF levels are observed in AD. Therefore, NPTX2 might not be suited for clinical diagnostic purposes or to distinguish different mechanisms causing pathological synaptic dysfunction and loss. Yet, for future mechanistic studies it needs to be considered that these markers could be indicative of different pathological processes specific for either DLB or AD. In the former, we hypothesize here that decreased levels might be considered in relation to GABAergic interneuron of the visual cortex. In the later, studies have shown a relations between NPTX2 and hippocampal volume and functional connectivity in the salient/attention networks[1, 2].
In conclusion, decreased CSF levels of NPTX2 might be potent biomarker for indicative of a specific neurodegenerative processes in DLB related to the visual spatial domain. In both DLB and AD, a possible explanation for these decreased levels, based on literature, suggest that this process might be driven by decreased signaling via the VGF/BDNF pathway. Further understanding of how these processes intertwine and how they are reflected in CSF could give insight into the neurodegenerative processes occurring in both DLB and how these might differ from AD. possibly aiding in early diagnosis.”
Point 2 reviewer: In order to be able to compare with AD, it is necessary that the authors include the same analysis showed in Fig 1D (ROC analysis) and Table 2 with the data obtained from AD CSF.
Response:
At the request of the reviewer we have added the ROC curve analysis for both the comparisons between SCD and AD and that between DLB and AD. The latter also links to the previous comment and shows that these markers indeed have limited capacity to distinguish AD and DLB. We have performed the same analyses that were performed for DLB patients on the neuropsychological test data of the AD patients. Yet, these had a low power due to the small sample size.
We have added the following text to the results and discussion sections.
Results: Lines 240-248
“Similar analyses were performed comparing AD to SCD and AD to DLB. Binary logistic regression models were run for each marker independently and all markers combined, including age as covariate in all analysis. ROC curve analysis was performed for both comparisons and is shown in supplemental Figure S9. The model with all three markers combined had the highest sensitivity and specificity differentiating AD and SCD (combined markers AUC=0.963). This was mostly driven by NPTX2 (AUCS=0.795) and α-synuclein (AUC=0.782) levels, which was reflected in their individual ROC curves being the only two significantly different from 0.5. Limited sensitivity and specificity were observed for these markers when differentiating between AD and DLB (combined markers AUC=0.759).”
Results: Lines 275-280
“Relations between NPTX2 CSF levels and cognitive function and decline were examined for AD patients using similar models as applied for the DLB patients. Data are shown in Supplemental Table S10. No significant relations between NPTX2 level and cognitive function or decline were observed in the AD group. For both the MMSE and Global cognitive score, 95% confidence intervals indicate a possible positive relation between NPTX2 levels and the cognition, however, these were not significant due to limited power in this group.”
Discussion: lines 297-300
“These findings were not observed in the current study, which was probably due to limited number of AD patients compared to these previous studies, especially for the evaluation of cognitive decline during follow-up. However, a trend for the relation between the MMSE and global cognitive score was observed.”
Reviewer 2 Report
In the manuscript entitled “Pathologically decreased CSF NPTX2 levels in DLB are related to the synaptic marker’s alpha-synuclein and VGF,” Walten and colleagues examine the levels of synaptic protein marker NPTX2 in the CSF of DLB patients. And they compare NPTX2 levels to the levels of other synaptic proteins VGF and α-synuclein. Further, the authors attempt to relate NPTX2 levels to cognitive test scores. The authors report the decreased levels of NPTX2 in the CSF of DLB patients as compared to cognitively healthy individuals. Interestingly, they show that the changes in the levels of CSF NPTX2 are closely associated with synaptic dysfunction and loss in DLB. The current study extends their recent publication wherein they identified 69 proteins, which were differentially expressed in DLB, as compared to controls. In the same study, they showed the low levels of CSF NPTX2 and VGF in DLB patients, which were associated with cognitive decline. However, in the current work, the authors explore the interdependence of these synaptic biomarkers using a decent number of CSF samples. Although the research topic is relevant for the readers of this journal, the inclusion of CSF samples from other disease groups could be important to support the authors' claim. Here, I have some concerns and suggestions.
Concerns:
- Since the levels of NPTX2, VGF and alpha-synuclein appear to be highly interdependent. Under pathological conditions, the changes in the levels of alpha-synuclein due to aggregation result in the disbalance of other biomarkers (NPTX2 and VGF). Thus, it is necessary to analyze the effects of alpha-synuclein aggregation on the levels of NPTX2 and VGF biomarkers in the CSF samples from PD and PDD patients. Whether a similar result will be observed as seen with DLB patients or different.
- The low levels of NPTX2 have been observed in the CSF of both DLB and AD; however, it is not very clear how the low levels of NPTX2 in these diseases lead to two different cognitive fates. Please elaborate.
- Please check the asterisk for the corresponding author.
- In lane 329, please correct the spelling of “Marker”
- In Suppl.FigureS5 C, please correct “NPTX2 VS VGF”
Author Response
Rebuttal addressing the points of the reviewer
We want to thank the reviewer for the thorough examination of the manuscript and the interest in previous publication by our group on the subject. We hope that we have sufficiently addressed the concerns brought-up by the reviewer.
Concern 1 reviewer: Since the levels of NPTX2, VGF and alpha-synuclein appear to be highly interdependent. Under pathological conditions, the changes in the levels of alpha-synuclein due to aggregation result in the disbalance of other biomarkers (NPTX2 and VGF). Thus, it is necessary to analyze the effects of alpha-synuclein aggregation on the levels of NPTX2 and VGF biomarkers in the CSF samples from PD and PDD patients. Whether a similar result will be observed as seen with DLB patients or different.
Response:
It is our understanding that the reviewer suggests that the analysis of VGF and NPTX2 should be extended into other α-synucleinopathies to examine if their interdependence can also be observed within these pathologies. Unfortunately, we are not able to perform these additional studies within the timeframe for these revisions. Examining these markers would require analysis in a novel cohort and would be outside of the possibilities for these study, as these samples are not readily available for additional analysis and would require a new ethical committee approval, sample retrieval and lab-analysis, which is even more complicated now due to the pandemic. We do believe that sufficient novel data is being presented with a decent number of individuals as indicated by the reviewer, with which we show that CSF NPTX2 levels are decreased, correlates to other synaptic markers, and are related to cognitive decline in the visual spatial domain. We do believe that examining the role of alterations in these specific synaptic markers might be of interest for future studies to examine if the principle neuron-interneuron inhibitory circuit is affected.
Concern 2 reviewer: The low levels of NPTX2 have been observed in the CSF of both DLB and AD; however, it is not very clear how the low levels of NPTX2 in these diseases lead to two different cognitive fates. Please elaborate.
Response:
We argue that the lower levels of NPTX2 are not casual but consequential to synaptic loss and thus the different pathological processes indeed lead to both lower CSF values and different cognitive fates. To that account, the biomarkers described here have limited capacity to distinguish these pathological processes and thus limited use as a diagnostic tools. The lower NPTX2 levels in both AD and DLB might be related to different pathological processes and resemble synaptic loss in different brain regions. To that extend some data has been published on the role of GABAergic interneurons of the visual cortex in DLB, discussed in lines 300-314 of the manuscript:
“In the data presented, we observed reduced NPTX2 levels in CSF of DLB patients, comparable to those observed in AD. NPTX2 functions as signal regulator in the excitatory synapses between principal neuron and GABAergic interneurons [3] and activity and number of these synaptic connections would impact NPTX2 levels in CSF. Lower NPTX2 levels would thus be an indication of lower excitation of inhibitory neuronal networks and thereby decreased GABAergic inhibitory signaling. This process has been related to memory decline and hippocampal volume in AD [2] and the changes observed in DLB might also relate to these processes. However, memory and hippocampal volume are less prone to suffer from degeneration in DLB [4, 5]. We observed a prominent relation between NPTX2 levels and visual spatial decline. Interestingly, the GABAergic interneurons of the visual cortex have been implicated in DLB [6] and decline of synaptic functions in this region is also observed in DLB [7]. Together, these findings suggest that reduced levels of NPTX2 might play a critical role in visual problems commonly observed in DLB patient by interfering in the principal neuron-interneuron circuitry. Yet, further validation studies of NPTX2 expression in post-mortem tissue of DLB patients and validation in another DLB cohort would be required to confirm this hypothesis”
That we observe a relation between decreased performance in the visual spatial domain and lower NPTX2 might be indicative of this process. In AD, studies have shown a relations between NPTX2 and hippocampal volume and functional connectivity in the salient/attention networks[1, 2]. However, we cannot confirm the hypothesis that these are different mechanisms based of on CSF data, due to lack in spatial specificity. We adapted the manuscript to include this limitation into the discussion and have adjusted our conclusion accordingly:
Discussion: lines 358 line 365
“Although CSF levels of NPTX2 were deceased in DLB, similar changes in CSF levels are observed in AD. Therefore, NPTX2 might not be suited for clinical diagnostic purposes or to distinguish different mechanisms causing pathological synaptic dysfunction and loss. Yet, for future mechanistic studies it needs to be considered that these markers could be indicative of different pathological processes specific for either DLB or AD. In the former, we hypothesize here that decreased levels might be considered in relation to GABAergic interneuron of the visual cortex. In the later, studies have shown a relations between NPTX2 and hippocampal volume and functional connectivity in the salient/attention networks [1, 2].”
Concerns 3,4 5
Response:
We thank the reviewer for pointing out these inaccuracies and they have been adapted accordingly.
- Soldan, A., et al., Resting-State Functional Connectivity Is Associated With Cerebrospinal Fluid Levels of the Synaptic Protein NPTX2 in Non-demented Older Adults. Front Aging Neurosci, 2019. 11: p. 132.
- Xiao, M.F., et al., NPTX2 and cognitive dysfunction in Alzheimer's Disease. Elife, 2017. 6.
- Chang, M.C., et al., Narp regulates homeostatic scaling of excitatory synapses on parvalbumin-expressing interneurons. Nat Neurosci, 2010. 13(9): p. 1090-7.
- Walker, L., et al., Neuropathologically mixed Alzheimer's and Lewy body disease: burden of pathological protein aggregates differs between clinical phenotypes. Acta Neuropathol, 2015. 129(5): p. 729-48.
- McKeith, I.G., et al., Diagnosis and management of dementia with Lewy bodies: Fourth consensus report of the DLB Consortium. Neurology, 2017. 89(1): p. 88-100.
- Khundakar, A.A., et al., Analysis of primary visual cortex in dementia with Lewy bodies indicates GABAergic involvement associated with recurrent complex visual hallucinations. Acta Neuropathol Commun, 2016. 4(1): p. 66.
- Mukaetova-Ladinska, E.B., et al., Synaptic proteins and choline acetyltransferase loss in visual cortex in dementia with Lewy bodies. J Neuropathol Exp Neurol, 2013. 72(1): p. 53-60.
Round 2
Reviewer 1 Report
Authors have addressed all the question raised previously. No further comments.
Author Response
No further comments.
Reviewer 2 Report
I have no further comments.
Author Response
No further comments